# Optimizing Somatic Embryogenesis Initiation, Maturation and Preculturing for Cryopreservation in *Picea pungens*

Xi Cao [1,2,†], Fang Gao [2,†], Caiyun Qin [2], Shigang Chen [2], Jufeng Cai [2], Changbin Sun [3], Yuhui Weng [4,*] and Jing Tao [2,*]

1   College of Horticulture, Jilin Agricultural University, 2888 Xincheng St., Changchun 130118, China
2   Jilin Provincial Academy of Forestry Sciences, 3528 Linhe St., Changchun 130033, China
3   Changchun Academy of Forestry, 5840 Jingyue St., Changchun 130117, China
4   Arthur Temple College of Forestry and Agriculture, Stephen F. Austin State University, Nacogdoches, TX 75965, USA
*   Correspondence: wengy@sfasu.edu (Y.W.); taojing8116@126.com (J.T.)
†   The authors contributed equally to this work.

**Abstract:** *Picea pungens* (Engelm.), known for its blue-green needles, has become a likable ornamental species in northeast China since 2000. Nonetheless, a lack of propagation methods that can maintain genetic fidelity and develop seedlings at a large scale prevents the further expansion of the species. Somatic embryogenesis (SE), paired with cryopreservation technologies, may provide a valid alternative. *Picea pungens* SE is not new, but its practical application has been limited due to low efficiencies in SE initiation and maturation as well as a lack of effective cryopreservation technology. In this study, experiments were carried out to overcome the limitations by modifying culture media. For initiation, the efficiency was enhanced by adjusting concentrations of 2.4-dichlorophenoxy acetic acid (2,4-D), 6-benzyl amino–purine (6-BA) or sucrose supplemented to the induction medium. The concentrations of 4.0 mg/L 2,4-D, 2 mg/L 6-BA, and 5 to 10 g/L sucrose were found optimal in maximizing initiation efficiency. For maturation, the efficiency, expressed as the number of mature somatic embryos per gram of fresh mass cultured (E/gFM), varied greatly with the choices of the basal medium and concentration of abscisic acid (ABA) of the maturation medium. Based on our results, the judicial choices were using the DCR medium as the basal medium and 10 mg/L ABA. The maturation efficiency could also be improved by adjusting the maturation medium's osmotic pressure by manipulating the concentrations of carbohydrate and Gelrite and culture density. While the maturation medium, using sucrose as carbohydrate source or supplemented with a low (<8 g/L) Gelrite concentration, facilitated maturation, optimal selections were truly genotype-dependent. Our results also suggest that, while the optimal culture density varied with genotype, in general it is needless to culture more than 100 mg embryogenesis tissues per dish (size: $10 \times 1.5$ cm). Based on this study, the optimum pretreatment for embryogenesis tissue cryopreservation was culturing the tissues on the proliferation medium with 0.4 mol/L sorbitol for 24 h, followed by treatment with 5% Dimethyl sulfoxide. This study significantly improved the initiation (achieved a frequency of 0.56) and embryo maturation efficiencies (achieved 1030 E/gFM) and established an effective preculturing protocol for cryopreservation (recovered 1354 E/gFM) for the species. The protocols developed here, paired with the available ones for other SE steps in the literature, form a well-refined SE technology intended for commercial application to *Picea pungens*.

**Keywords:** *Picea pungens*; somatic embryogenesis; initiation; maturation; cryopreservation

## 1. Introduction

*Picea pungens* (Engelm.) is a well-known ornamental species, primarily due to its beautiful blue-green needles, in North America and Europe. China has been introducing seeds and seedlings of the species from the United States since 2000, and now it has become

a desired landscaping species, particularly in northeast China. The demand for planting the species has been increasing over the past 20 years. At present, planting materials are mainly derived from seed propagation. To meet seed demand, a genetic selection program has been initiated, and seed production areas have been established in northeast China [1]. Nonetheless, seed propagation has difficulty in retaining desirable parental tree characteristics, i.e., needle color in particular [1], a barrier for further expansion of the species. Developing propagation methods, which cannot only produce seedlings rapidly at a large-scale but can also maintain genetic fidelity, is ideal. Somatic embryogenesis (SE) is believed to be a valid alternative.

Somatic embryogenesis in conifers is a recently developed vegetative propagation technique through which an unlimited number of genetically identical copies of trees can be produced from a single seed. Relative to other vegetative propagation methods (i.e., rooting, cutting and grafting), SE has advantages of having a higher propagation rate, less impact from the donor plant's age and physiological state, less seasonal restrictions for production and greater amenability to the long-term storage of tissues [2]. However, SE is an integrated process consisting of multiple, sequential steps, progressing through initiation, proliferation, maturation, germination and acclimation. For a practical SE program, a sufficient efficiency is indispensable for each step.

Previous work on SE in *Picea pungens* has shown the potential of integrating the technology into operational seedling production. Afele [3] first reported the success of SE initiation, proliferation and maturation of the species, which was further improved by the subsequent work [4,5]. Most recently, Tao [1] achieved satisfactory efficiencies in SE proliferation, germination and acclimation. Despite the advancements, efficiencies in two early SE steps—the embryogenic tissue (ET) initiation (frequency ranged from 20%~30%) and maturation (the best efficiency was 188 mature embryos per gram ET cultured)— although sufficient to generate embryogenic cultures for experimentation, are insufficient in terms of commercial application of the technology. An initiation frequency of 50% or higher [6,7] and a maturation efficiency of generating 300 or more mature embryos per gram of fresh mass cultured have been achieved [8] in *Picea glauca*, *Picea mariana* and *Picea abies*, the most highly commercialized SE programs so far. It is apparent that improvements in initiation and maturation are necessary before commercializing SE technology in *Picea pungens*. While *Picea pungens* SE initiation and maturation responded to changes in many factors, their optimum choices to maximize efficiencies have not been closely examined. Among the factors, the conditions of culture media, in particular the concentrations of plant growth regulators (PGRs) and medium osmotic pressure, are critical to initiation and maturation [1,3–5].

The most important application of SE is probably in conjunction with genetic selection programs [2]. Through controlled breeding, new genotypes may be developed. ETs of the genotypes may be developed using the SE technology; part of the ETs are used to develop seedlings for genetic tests and others for cryopreservation. Genetically superior genotypes may be identified in genetic tests often 5 to 10 years post-establishment; identical trees can then be regenerated via SE using cryopreserved ETs of the selected genotypes for commercial purpose [2,9,10]. Thus, an effective cryopreservation technique is essential so that ETs can be preserved, without losing genetic integrity and viability during the testing period. Most current cryopreservation protocols entail a critical preculturing process: the incubation of ETs in a proliferation medium of decreased osmotic potential, treated with osmotically active compounds (i.e., sorbitol, sucrose) and cell cryoprotectants, such as dimethylsulfoxide (DMSO). In conifer, improving plant development from cryopreserved ETs through optimization of the culture media and environmental conditions has been reported [11]. Determination of suitable preculturing conditions, including culture period and optimal concentrations of sorbitol and DMSO, is critical for achieving a high cryopreservation efficiency [11], which, however, is often species- and/or genotype-dependent [11–13]. To our knowledge, no study regarding cryopreservation in *Picea pungens* has been reported, limiting the full utilization of advantages of the SE technology and genetic selection.

This study was divided into three main work areas, with an overall objective to overcome main limitations intended for commercializing *Picea pungens* SE technology. Specifically, the first goal was to improve ET initiation efficiency via modifying concentrations of PGRs and sucrose supplemented to the initiation medium. The second goal was to screen out the conditions suitable for maturation by modifying the maturation medium and culture density in culture. The last goal was to develop preculturing conditions for cryopreserving ETs via selecting suitable concentrations of sorbitol and DMSO along with preculture periods. We expected that the study would help the forest sector overcome the barriers and develop feasible and productive protocols for mass vegetative propagation of *Picea pungens*.

## 2. Materials and Methods

Long-term, stored ($-20\ ^\circ$C), mature seeds of *Picea pungens* of 3 provenances (F1, F2 and F3, bought from Carson Forest, NM, USA in 2010, from San Isabel, CO, USA in 2008 and Rio Grande Forest, CO, USA in 2008, respectively) were used as materials. Prior to embryo extraction, seeds were washed with running water for 18 h, followed by sterilizing with 75% alcohol for 30 s, rinsing with sterile water 3–5 times, sterilizing with NaClO solution (available 4% chlorine) for 15 min and finalized with rinsing 3–5 times with sterile water. Mature zygotic embryos were extracted from seeds under a microscope. Various experiments on improving initiation, maturation and cryopreservation were carried out. The details are presented in Table 1.

**Table 1.** Basal medium, medium supplements and treatments by experiment.

| Experiment | Basal Medium and Supplements | Treatment and Level |
|---|---|---|
| | Initiation | |
| Standard | mLV, sucrose 10 g/L, acid hydrolyzed casein 0.8 g/L, L-glutamine 0.5 g/L, Gelrite 4 g/L, 6-BA 2.0 mg/L, 2,4-D 4.0 mg/L | |
| Exp. 1.1 | Same as the standard but varying 2,4-D | 2,4-D: 0, 2.0, 4.0, 6.0 mg/L |
| Exp. 1.2 | Same as the standard but varying 6-BA | 6-BA: 0, 1.0, 2.0, 3.0, 4.0 mg/L |
| Exp. 1.3 | Same as standard but varying sucrose | Sucrose: 5, 10, 20, 30 g/L |
| | Maturation | |
| Standard | mLV, sucrose 30 g/L, acid hydrolyzed casein 0.8 mg/L, L-glutamine 0.5 g/L, Gelrite 6 g/L, active carbon 1 g/L, ABA 13.22 mg/L | |
| Exp. 2.1 | Same as standard but varying basal medium | Basal medium: DCR, mLV, 1/2 MS |
| Exp. 2.2 | Same as the standard but varying carbohydrate type and concentration | Carbohydrate: sucrose, glucose, maltose Concentrations: 15, 30, 45, 60 g/L |
| Exp. 2.3 | Same as the standard but varying Gelrite | Gelrite: 4, 6, 8, 10 g/L |
| Exp. 2.4 | Same as the standard but varying ABA | ABA: 0, 10, 20, 30, 40 mg/L |
| Exp. 2.5 | Same as the standard | Cell density: 50, 80, 110, 140, 170 mg/dish |
| | Cryopreservation | |
| Standard | Proliferation medium *, sorbitol 0.4 mol/L, preculture period 18 h, DMSO 7.5% | |
| Exp. 3.1 | Same as the standard but varying sorbitol and preculture period | Sorbitol: 0, 0.2, 0.4, 0.8 mg/L Preculture period: 0, 12, 18, 24, 36 h |
| Exp. 3.2 | Same as the standard but varying DMSO | DMSO: 0, 5, 7.5, 10, 15% |

* the proliferation medium can be found in Tao et al. (2021) [1].

### 2.1. Initiation of Embryogenic Tissues

The embryos of F1, F2 and F3 were used as explants. The initiation medium was a modified Litvay's medium (mLV) [14], supplemented with sucrose (10 g/L), acid hydrolyzed casein (0.8 g/L), L-glutamine (0.5 g/L), Gelrite (4 g/L), 6-benzylaminopurine (6-BA; 2.0 mg/L) and 2,4-dichlorophenoxyacetic acid (2,4-D; 4.0 mg/L) (referred to as the standard initiation medium) (Table 1). The pH of the medium was adjusted to 5.75–5.80 prior to sterilization.

In experiments, the concentration of one supplement (2,4-D, 6-BA or sucrose) in the standard initiation medium was varied while keeping others constant (Table 1). The concentrations of each supplement were specifically selected so that they covered the typic range likely to be used in spruce SE. For each experiment, a factorial design (provenance and treatment) was used. For each level of a treatment, 10 embryos were inoculated in a Petri dish, which was repeated 10 times (dishes). After the inoculation, the Petri dishes were kept in the dark at $23 \pm 2$ °C to initiate ETs, which were monitored using a stereo microscope (OLYMPUS SZX 7, Tokyo, Japan). The ET initiation was assessed at 60 days post-culture by recording the success or failure of each cultured embryo. Success was defined for initiation of an embryo if ETs were established. Once an embryo had developed ETs, it was given a clone line number.

*Exp. 1.1: Effect of 2,4-D concentration*

The concentration of 2,4-D in the standard initiation medium was varied. Four 2,4-D concentrations (0, 2.0, 4.0 and 6.0 mg/L) were tested.

*Exp. 1.2: Effect of 6-BA concentration*

The concentration of 6-BA in the standard initiation medium was varied. Five concentrations of 6-BA (0, 1.0, 2.0, 3.0 and 4.0 mg/L) were tested.

*Exp. 1.3: Effect of sucrose concentration*

The concentration of sucrose supplemented to the standard initiation medium was varied. Four sucrose concentrations (5, 10, 20 and 30 g/L) were tested.

## 2.2. Maturation of Somatic Embryos

The inducted ETs on the standard initiation medium were proliferated by clone line in a proliferation medium, which was the same as the standard initiation medium but with a PGR combination of 2.0 mg/L 2,4-D and 1.0 mg/L 6-BA. The ETs were cultured in the dark at $23 \pm 2$ °C and sub-cultured on the fresh proliferation medium once every 14 days.

Proliferated fresh ETs of three clone lines (1–9 was derived from F1, 2–22 and 2–23 from F2) were used as materials. For each line, approximately 80 mg of fresh ETs were added into a centrifuge tube containing 3 mL liquid, PGR-free proliferation medium. The tube was vigorously agitated to break up the tissues into a fine suspension, which was then poured over a sterile filter paper (Fushun City Civil Affairs Filter Paper Factory, Liaoning, China) to remove liquid. The filter paper, with cells attached, was transferred onto a dish containing fresh and solid maturation medium. The standard maturation medium used the mLV as the basal medium, supplemented with 30 g/L sucrose, 0.8 g/L acid hydrolyzed casein, 0.5 g/L L-glutamine, 6 g/L Gelrite, 1 g/L active carbon and 13.22 mg/L abscisic acid (ABA) (Table 1).

In the maturation experiments, one component (basal medium, osmoticum (via carbohydrate and Gelrite) and ABA) of the standard maturation medium was varied while keeping all others constant (Table 1). The concentrations of each component were selected based on previous spruce SE studies. Also, an experiment varying culture density was conducted to examine effects of density. Each experiment was carried out using a factorial design, and each treatment level was repeated six times (dishes). Culture was fulfilled in the dark at $23 \pm 2$ °C. After 60 days culture, the number of mature embryos was counted; that number was then expanded to a per gram fresh mass cultured (E/gFM) unit as an indicator of maturation efficiency.

*Exp. 2.1: Effect of basal medium*

The basal medium of the standard maturation medium was varied. Three basal media—the DCR [15], mLV and half-strength Murashige-Skoog (1/2 MS) [16]—were tested.

*Exp. 2.2: Effects of carbohydrate type and concentration*

The carbohydrate type and its concentration supplemented to the standard maturation medium were varied. Three different types (sucrose, glucose and maltose)—each paired with four concentrations (15, 30, 45 and 60 g/L)—were tested.

*Exp. 2.3: Effect of Gelrite concentration*

The concentration of Gelrite in the standard maturation medium was varied. Four Gelrite concentrations (4, 6, 8 and 10 g/L) were tested.

*Exp. 2.4: Effect of ABA*

The dose of ABA, supplemented to the standard maturation medium, was varied. Five concentrations of ABA (0, 10, 20, 30 and 40 mg/L) were tested.

*Exp. 2.5: Effect of cell density*

The standard maturation medium was used. The cell density cultured in a dish (size: $10 \times 1.5$ cm) was varied. Five culture densities (50, 80, 110, 140 and 170 mg ETs per dish) were tested.

### 2.3. Cryopreservation of Embryogenic Tissues

Cryopreservation experiments were carried out with clone line 1–8, which was derived from F1. The standard procedure for preculturing ETs before cryopreservation was as follows: 1.5 g fresh ETs were cultured on the proliferation medium for 7–10 days, which were then transferred into a sterile conical flask containing 6.38 mL liquid proliferation medium paired with 0.4 mol/L sorbitol. The flask was shaken at a speed of 120 rpm for 18 h (referred to as the preculture period) in a condition of dark and $23 \pm 2$ °C temperature. Dimethylsulfoxide (DMSO) was filter sterilized and then added to the flask at a 7.5% concentration. The flask was kept at an ice-water mixture for 90 min. After that, the liquid was dispensed into 1.5 mL sterile cryovials. The cryovials were transferred into a Nalgene program cooling box (Lincolnshire, NE, USA) of 4 °C, which gradually reduced to −80 °C, and the cryovials were kept for 2 h. The cryovials were then transferred to liquid nitrogen for storage. Experiments varying preculturing conditions (a combination of sorbitol concentration and preculture period, and DMSO concentration) were carried out while maintaining others constant (Table 1).

*Exp. 3.1: Effects of sorbitol concentration and preculture period*

The sorbitol concentration and preculture period of the standard preculturing procedure were varied. Combinations of four concentrations of sorbitol (0, 0.2, 0.4 and 0.8 mol/L) and five preculture periods (0, 12, 18, 24 and 36 h) were tested based on a factorial design, with each combination being repeated 6 times.

*Exp. 3.2: Effect of DMSO on ET cryopreservation*

The DMSO concentration of the standard preculturing procedure was varied. Five DMSO concentrations (0, 5, 7.5, 10 and 15%) were tested based on a completely randomized design, with each level being repeated 6 times.

After 30 days of cryopreservation, the efficiency of cryopreservation was evaluated by culturing the cryopreserved ETs. ETs were thawed in a sterile water bath at 37 °C for 1 min and then transferred to a Buchner funnel with sterile filter paper to remove the cryoprotectant. The filter paper with ETs was transferred to a dish with the proliferation medium for 1 day and then transferred to another dish with the same but fresh proliferation medium. The ETs were cultured in the dark at $23 \pm 2$ °C for 15 days for recovery. The recovered ETs (about 80 mg per dish) were cultured on the maturation medium, and the number of mature embryos was counted after 60 days and expanded to a per gram fresh mass basis.

### 2.4. Data Analysis

In the initiation experiments, the response variable, the number of successes out of the total number of embryos cultured (a data in the 'event/trial' type), is assumed to follow a binomial distribution, and the data were analyzed following the model (1):

$$y_{ijk} = \mu + F_i + T_j + FT_{ij} + \varepsilon_{ijk} \tag{1}$$

where $y_{ijk}$ is the success or failure in initiation of the $k$th embryo of the $i$th provenance and $j$th level of a treatment (2,4-D, 6-BA or sucrose); $\mu$ is the overall mean; $F_i$, $T_i$ and $FT_{ij}$ are the fixed effects of the provenance, treatment and the interaction between them, respectively; and $\varepsilon_{ijkl}$ is the random error, which is assumed NID $(0, \sigma^2)$, with $\sigma^2$ being error variance.

In the maturation experiments, the maturation efficiency, the total number of mature embryos per gram fresh mass cultured, is count data and was modeled using the model [2] paired with a negative binomial distribution (similar to Poisson distribution but can handle the over-dispersion issue) for all experiments other than Exp. 2.2:

$$y_{ijk} = \mu + L_i + T_j + LT_{ij} + \varepsilon_{ijk} \qquad (2)$$

where $y_{ijk}$ is the number of mature embryos per gram fresh mass cultured; $L_i$ is the fixed effect of ith clone line; and all others are the same as described in the model (1). Model (2) was also used to analyze the data of Exp. 2.2, but an additional effect, the concentration within a carbohydrate type, was added.

The cryopreservation experiments had the same data type as the maturation experiments. The data of Exp. 3.1 were analyzed using the model (3) paired with a negative binomial distribution:

$$y_{ijk} = \mu + S_i + P_j + SP_{ij} + \varepsilon_{ijk} \qquad (3)$$

where $S_i$, $P_j$ and $SP_{ij}$ are the fixed effects of the sorbitol, preculture period and the interaction between them, respectively, and other terms were previously described. For Exp. 3.2, we used a model including only one factor, the DMSO treatment.

Statistical Analysis System (SAS) programs [17] were used in the analyses, i.e., GEN-MOD procedure for the "event/trial" data and GLIMMIX procedure for the count data. Means and their standard errors (SE) were computed either on an original scale directly or transformed to the original scale via a link, depending on data type. Multiple comparisons among levels of a treatment were made using the Tukey method. Unless otherwise stated, the significance level in this paper refers to $\alpha$ = 0.05.

## 3. Results

### 3.1. Initiation of Embryogenic Tissue

In Exp. 1.1, varying the concentration of 2, 4-D in the initiation medium affected the morphology of embryogenic cells (Figure 1) and initiation efficiency significantly (Table 2). A high portion of the tissues were browning (Figure 1a), and the frequency was low (0.06) on the 2,4-D-free medium; the induced tissues changed to filamentous and transparent in appearance after adding 2,4-D (Figure 1b–d), and the efficiency enhanced steadily with progressively adding 2,4-D to the medium, peaked at 4.0 mg/L achieving a frequency of 0.56 and thereafter dropped (Figure 2a). Compared to the 4.0 mg/L, the 2 and 6 mg/L had 33% and 41% lower frequencies, respectively. The frequency also differed significantly with provenance; averaged across 2,4-D levels, F1 and F3 had comparable frequencies (around 0.32), which were significantly higher than that of F2 (0.22) (Table 3). All provenances responded similarly to the 2,4-D concentration, as supported by the negligible interaction between the provenance and 2,4-D treatment (Table 2).

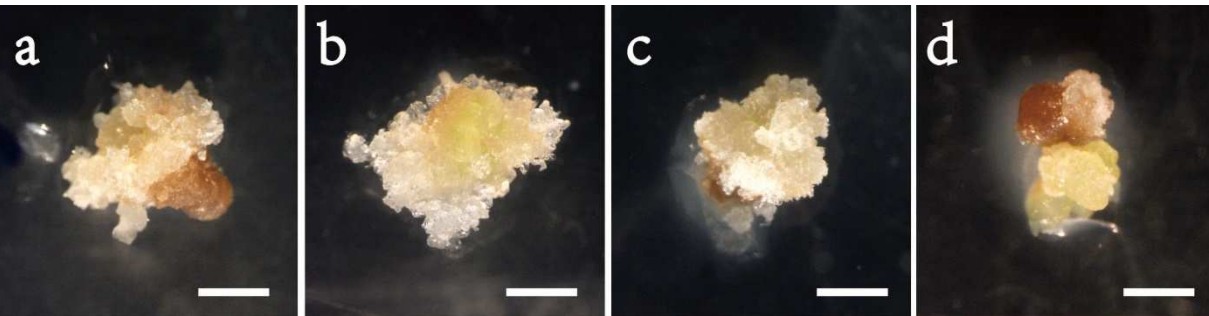

**Figure 1.** Initiation of embryogenic cell mass of the provenance F1 in *Picea pungens* SE under various 2,4-D concentrations: (**a**) 0 mg/L; (**b**) 2 mg/L; (**c**) 4 mg/L; and (**d**) 6 mg/L. Bars: (**a–d**) = 1 cm.

**Table 2.** Results of analysis of variance in *Picea pungens* somatic embryogenesis initiation experiments.

| Experiment | Source of Variation | df | F | Pr > F |
|---|---|---|---|---|
| Exp. 1.1 | 2,4-D | 3 | 191.33 | <0.0001 |
| | Provenance | 2 | 6.73 | 0.0345 |
| | 2,4-D X Provenance | 6 | 3.29 | 0.7718 |
| Exp. 1.2 | 6-BA | 4 | 151.73 | <0.0001 |
| | Provenance | 2 | 11.96 | 0.0025 |
| | 6-BA X Provenance | 8 | 9.74 | 0.2836 |
| Exp. 1.3 | Sucrose | 3 | 32.19 | <0.0001 |
| | Provenance | 2 | 22.63 | <0.0001 |
| | Sucrose X Provenance | 8 | 8.13 | 0.2288 |

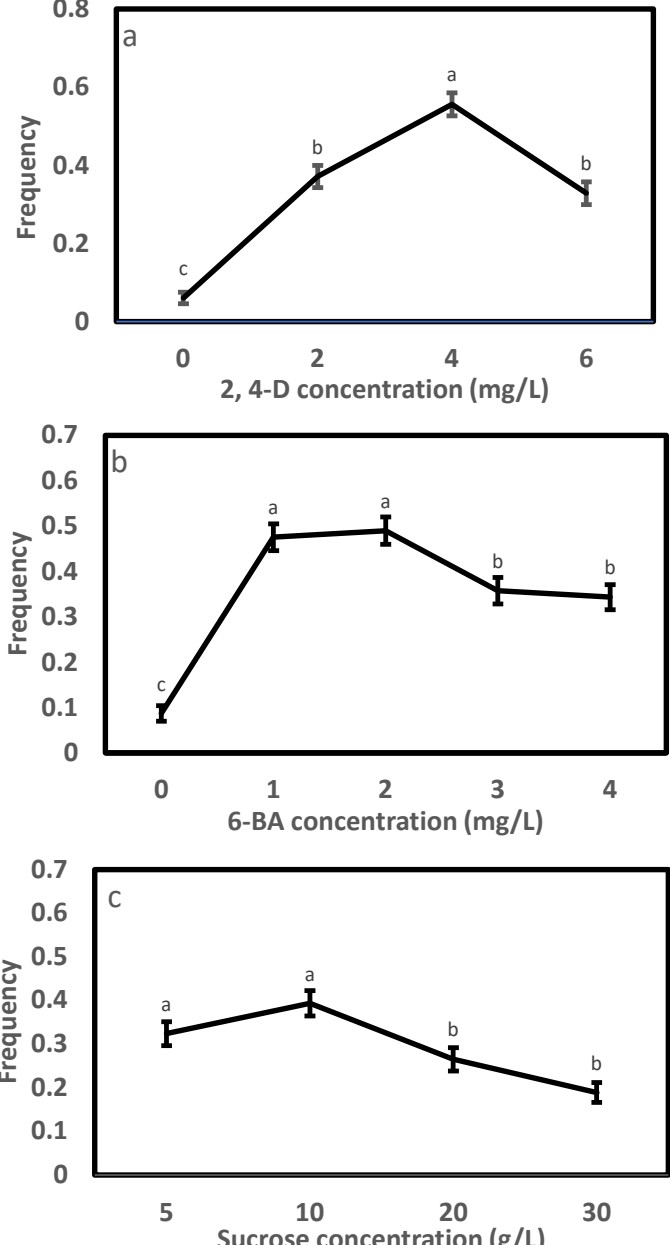

**Figure 2.** Initiation frequency and standard error by experiment: (**a**) 2,4-D, (**b**) 6-BA and (**c**) sucrose. Note that values with the same letters marked on a panel were statistically insignificant.

**Table 3.** The average initiation frequency by provenance with standard error in brackets obtained from *Picea pungens* initiation experiments.

| Provenance | Exp. 1.1 | Exp. 1.2 | Exp. 1.3 |
|:---:|:---:|:---:|:---:|
| F1 | 0.32 (0.03) [a] | 0.39 (0.02) [a] | 0.20 (0.02) [b] |
| F2 | 0.22 (0.03) [b] | 0.32 (0.03) [ab] | 0.34 (0.02) [a] |
| F3 | 0.31 (0.03) [a] | 0.27 (0.02) [b] | 0.33 (0.02) [a] |

Note: Mean values marked with the same letters in a column are not significantly different at $p < 0.05$.

The results of Exp. 1.2 (Table 2) were similar to those from Exp. 1.1. Embryos responded strongly in initiation with changing 6-BA concentration (Figures 2b and 3). Without adding 6-BA to the medium, the induced tissues were mainly browning and non-embryonic callus (Figure 3a), paired with a low frequency of 0.09. White, soft and filamentous ETs appeared in media supplemented with 6-BA (Figure 3b–e), and the highest frequency (0.49) was recorded at 1 and 2 mg/L. Compared to the 2 mg/L, the 3 or 4 mg/L reduced frequency by 27% or 30%. The frequency varied greatly with provenance, with F1 having the highest (0.39), followed by F2 (0.31) and F3 (0.27) (Table 3). The interaction between the provenance and 6-BA treatment was not significant (Table 2), and thus, a 1~2 mg/L 6-BA was the optimum for the SE initiation of all provenances.

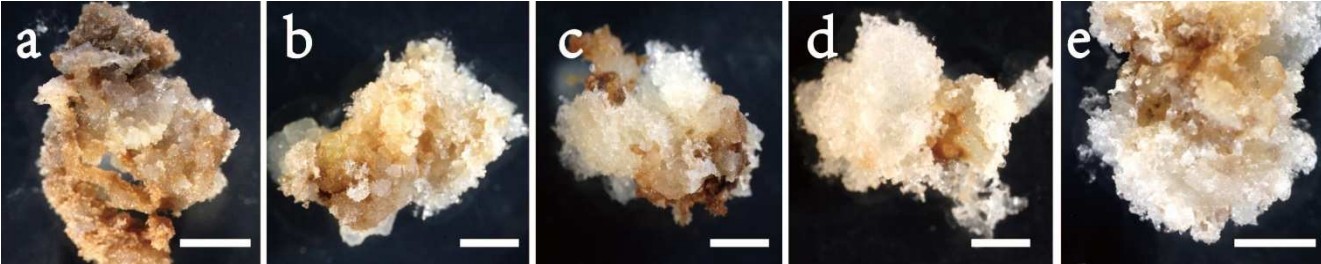

**Figure 3.** Initiation of embryogenic cell mass of provenance F1 in *Picea pungens* SE under various 6-BA concentrations: (**a**) 0 mg/L; (**b**) 1 mg/L; (**c**) 2 mg/L; (**d**) 3 mg/L; and (**e**) 4 mg/L. Bars: (**a**–**e**) = 1 cm.

The initiation frequency in Exp. 1.3 varied significantly with sucrose concentration as well as provenance but not their interaction (Table 2). Similar to Exp. 1.1 and 1.2, a pattern of "rise-peak-drop" in frequency with increasing sucrose concentration stood out (Figure 2c), with the highest frequency (0.39) being recorded at 10 g/L. Relative to the 10 g/L, a statistically comparable frequency was also achieved at 5 g/L sucrose (0.32), but those at the 20 and 30 mg/L were significantly lower by 33% and 52%, respectively. Averaged across sucrose concentrations, F2 and F3 had similar frequencies (0.34), which were significantly higher than that of F1 (0.20). Overall, a sucrose concentration from 5 to 10 g/L was ideal for the SE induction of *Picea pungens*, which was not changed with provenance.

### 3.2. Somatic Embryo Maturation

In Exp. 2.1, the maturation efficiency varied significantly with basal medium and clone line but not their interaction (Table 4). Combined over lines, the DCR medium recorded 744 E/gFM, which was about 75% higher than that obtained for the other two media (Figure 4a). Line 2–23 had the highest efficiency having 800 E/gFM, followed by 2–22 (461 E/gFM) and 1–9 (383 E/gFM) (Table 5). Overall, the DCR medium was the optimal choice for all lines.

**Table 4.** Results of analysis of variance in *Picea pungens* somatic embryogenesis maturation experiments.

| Experiment | Source of Variation | NDf * | DDf * | F | Pr > F |
|---|---|---|---|---|---|
| Exp. 2.1 | Basal medium | 2 | 45 | 5.37 | 0.0081 |
| | Clone | 2 | 45 | 8.25 | 0.0009 |
| | Basal medium * Clone | 4 | 45 | 0.51 | 0.7257 |
| Exp. 2.2 | Carbohydrate type (CT) | 2 | 186 | 1.92 | 0.1500 |
| | Clone | 2 | 186 | 3.12 | 0.0466 |
| | Concentration within a CT | 9 | 186 | 5.93 | 0.0002 |
| | CT * Clone | 4 | 186 | 3.03 | 0.0021 |
| Exp. 2.3 | Gelrite | 3 | 59 | 3.43 | 0.0228 |
| | Clone | 2 | 59 | 43.44 | <0.0001 |
| | Gelrite * Clone | 6 | 59 | 6.66 | 0.0001 |
| Exp. 2.4 | ABA | 4 | 60 | 6.69 | 0.0002 |
| | Clone | 2 | 60 | 7.66 | 0.0011 |
| | ABA * Clone | 8 | 60 | 4.48 | 0.0003 |
| Exp. 2.5 | Density | 4 | 75 | 1.74 | 0.1506 |
| | Clone | 2 | 75 | 55.25 | <0.0001 |
| | Density * Clone | 8 | 75 | 3.24 | 0.0032 |
| Exp. 3.1 | Preculture period (PP) | 4 | 80 | 20.17 | <0.0001 |
| | Sorbitol | 3 | 80 | 48.96 | <0.0001 |
| | PP * Sorbitol | 12 | 80 | 6.38 | <0.0001 |
| Exp. 3.2 | DMSO | 4 | 20 | 4.34 | 0.0109 |

* NDf and DDf are the numerator and denominator degree of freedom, respectively.

**Table 5.** The average number of matured embryos by clone line with standard error in brackets obtained from *Picea pungens* maturation experiments.

| Clone Line | Exp. 2.1 | Exp. 2.2 | Exp. 2.3 | Exp. 2.4 | Exp. 2.5 |
|---|---|---|---|---|---|
| 1–9 | 383.43 (51.14) [b] | 244.97 (23.99) [a] | 189.91 (27.19) [b] | 388.06 (39.44) [a] | 216.23 (42.32) [b] |
| 2–22 | 461.05 (61.46) [b] | 231.77 (26.49) [a] | 313.4 (44.71) [a] | 135.34 (13.58) [b] | 52.65 (15.77) [c] |
| 2–23 | 799.90 (106.47) [a] | 253.13 (29.40) [a] | 414.66 (59.16) [a] | 456.69 (45.24) [a] | 558.68 (55.02) [a] |

Note: Mean values marked with the same letters in a column are not significantly different at $p < 0.05$.

In Exp. 2.2, the average number by carbohydrate type ranged from 92 (maltose) to 372 E/gFM (sucrose). The differences between carbohydrate types, however, were statistically insignificant (Table 4), mainly due to a large standard error associated with the average of maltose (data not shown). The efficiency of carbohydrate type truly depended on clone line (Table 4). Lines 1–9 and 2–22 matured 116% and 78% more embryos on the medium using sucrose than the respective ones using glucose, but line 2–23 favored glucose to sucrose, producing 24% more, although statistically insignificant, embryos (Figure 5a). Within a carbohydrate type, the efficiency showed a "rise-peak-drop" pattern with increasing concentration, and the 30 mg/L concentration had the highest efficiency, irrespective of type (Figure 4b). Differences among concentrations of sucrose were insignificant (Figure 4b); even so, the 30 g/L obtained 489 E/gFM, which was 18% or 73% higher than that of the 15 g/L or 45 g/L.

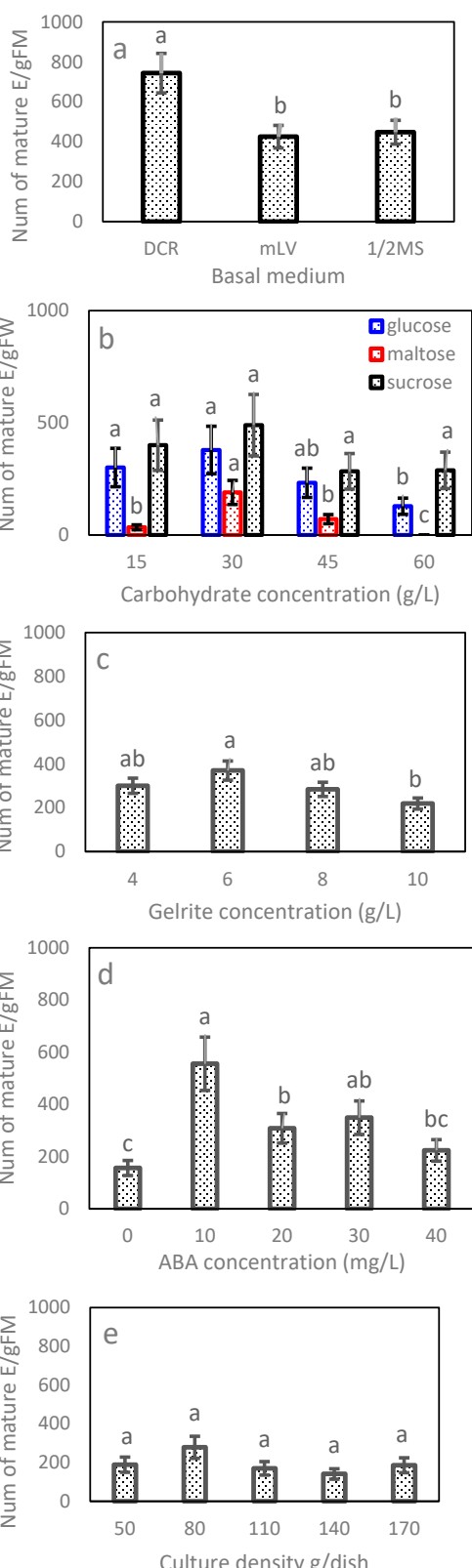

**Figure 4.** Maturation frequency and standard error by experiment: (**a**) basal medium, (**b**) carbohydrate type and concentration, (**c**) Gelrite, (**d**) ABA and (**e**) culture density. Values with the same letters marked on a panel were insignificant other than (**b**), of which values with the same letters marked within a carbohydrate type were insignificant. Note that E/gFM represents embryos obtained from per gram fresh weight tissues cultured.

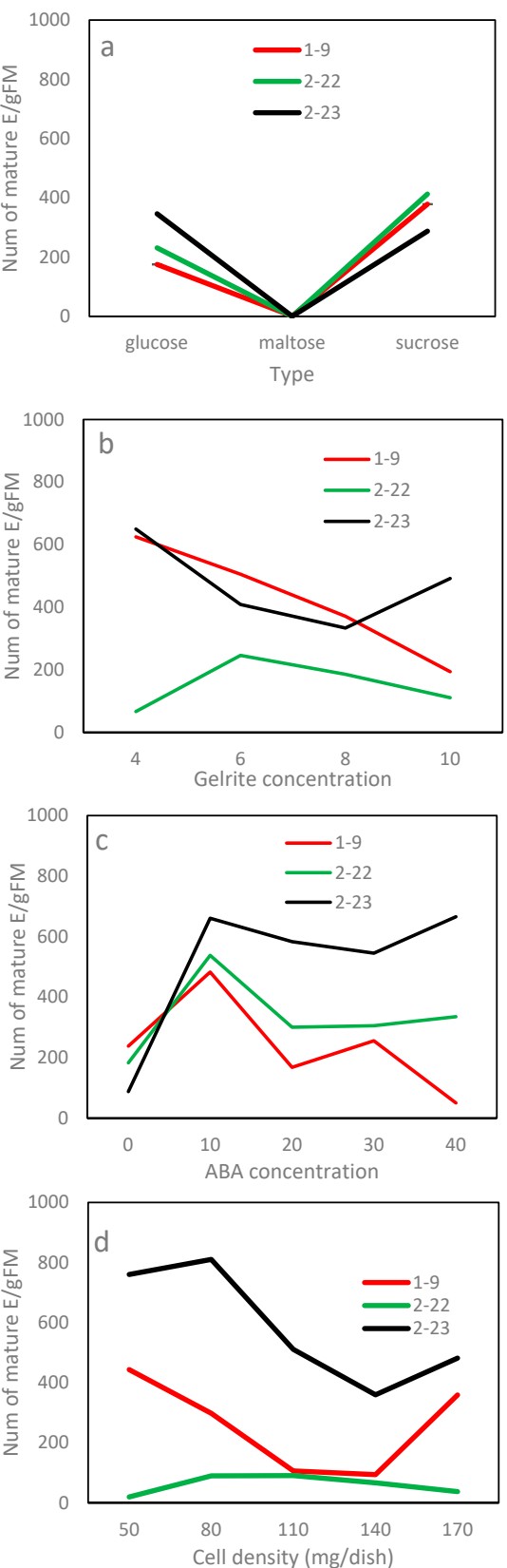

**Figure 5.** The interaction in maturation efficiency between the clone line and treatment by experiment: (**a**) carbohydrate type, (**b**) Gelrite, (**c**) ABA and (**d**) culture density. Note that E/gFM represents embryos obtained from per gram fresh weight tissues cultured.

For Exp. 2.3, the maturation efficiency varied substantially with the Gelrite concentration along with clone line and their interaction (Table 4). Combined over clone lines, the ET maturation responded in a pattern of "rise-peak-drop" to increasing Gelrite concentration (Figure 4c). The 6 g/L had the highest efficiency, while the 4 or 8 g/L achieved a statistically comparable, but lower, efficiency. The optimum concentration, however, was truly line-dependent (Figure 5b). Lines 2–23 and 1–9 had the highest efficiencies at the lowest concentration of 4 mg/L (620 and 625 E/gFM, respectively), while line 2–22 performed best at 6 mg/L (246 E/gFM).

In Exp. 2.4, mature embryos obtained in the maturation media of various concentrations of ABA varied, and ABA was needed for developing normal mature embryos (Figure 6). In terms of efficiency, Exp. 2.4 obtained similar results to those from Exp. 2.3 (Table 4). Combined over clone lines, the efficiency was low (156 E/gFM) on the ABA-free medium, improved with supplementing more ABA to the medium and dropped when ABA was overdosed (Figure 4d). The highest efficiency was recorded at 10 mg/L (560 E/gFM). By clone line, the efficiency ranged from 190 to 415 E/gFM, with the highest being for line 2–23. Even the interaction was significant; the 10 mg/L concentration appeared optimal for all clone lines (Figure 5c). Interestingly, for the two lines of F2, an increase was also observed when ABA concentration increased from 30 to 40 mg/L (Figure 5c).

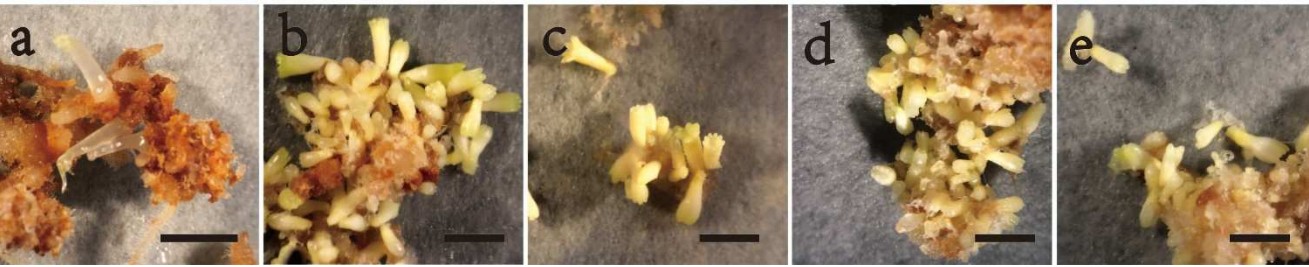

**Figure 6.** Mature somatic embryos of clone line 2–23 in *Picea pungens* SE under various ABA concentrations: (**a**) 0 mg/L; (**b**) 10 mg/L; (**c**) 20 mg/L; (**d**) 30 mg/L; and (**e**) 40 mg/L. Bars: (**a**–**e**) = 0.5 cm.

For Exp. 2.5, the differences among the culture densities were statistically insignificant (Table 4). Even so, averaged across lines, the 80 mg which had the highest efficiency obtained at least 48% more mature embryos than the other levels tested (Figure 4e). The efficiency varied with clone line along with the interaction between the line and density; therefore, the optimal culture density was line-dependent (Table 4; Figure 5d), optimizing at 80 mg for line 2–23 (810 E/gFM), 50 mg for 1–9 (443 E/gFM) and a wider range from 80 and 110 mg for 2–22.

### 3.3. Cryopreservation

In Exp. 3.1, the sorbitol concentration, preculture period and their interaction affected cryopreservation efficiency significantly (Table 4). Among the 20 combinations of the sorbitol concentration and preculture period, 9 successfully recovered mature embryos from cryopreserved tissues (Figure 7a). With preculturing on sorbitol-free media or having no preculture period, cryopreserved ETs failed to mature. The 0.2 mol/L sorbitol regenerated mature embryos only when the preculture period was 12 h (630 E/gFM). With increasing sorbitol concentration to 0.4 and 0.8 mol/L, embryos matured successfully under the preculture periods from 12 to 36 h. The best combination was 0.4 mol/L sorbitol paired with preculturing 24 h (1030 E/gFM). The combination of 0.8 mol/L sorbitol and 12 h culturing achieved a slightly lower, but statistically comparable, efficiency (918 E/gFM).

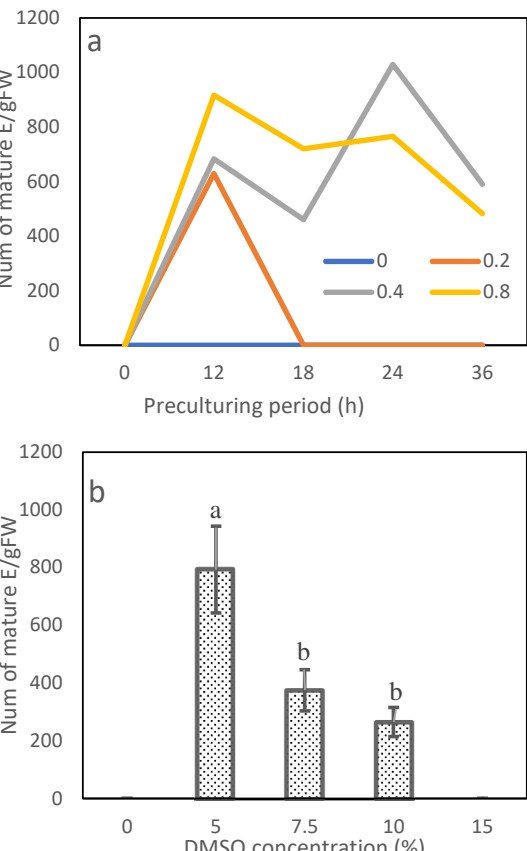

**Figure 7.** Maturation efficiency from cryopreserved embryo tissues by experiment: (**a**) combination of sorbitol concentration and preculture period and (**b**) DMSO. Values with the same letters marked on (**b**) were insignificant.

In Exp. 3.2, the DMSO concentration had a significant impact on the maturation efficiency (Table 4). Cryopreserved ETs failed in recovery and mature in the medium when no DMSO treatment was applied or when they were treated with the highest concentration, 15.0% DMSO (Figure 7b). The cryopreserved tissues were normally recovered, proliferated and developed into normal somatic seedlings when the tissues were pretreated with 5.0% or 7.5% DMSO (Figure 8). The highest efficiency was observed at 5.0% DMSO achieving 793 E/gFM, which was two times more than that of the second effective concentration of 7.5%.

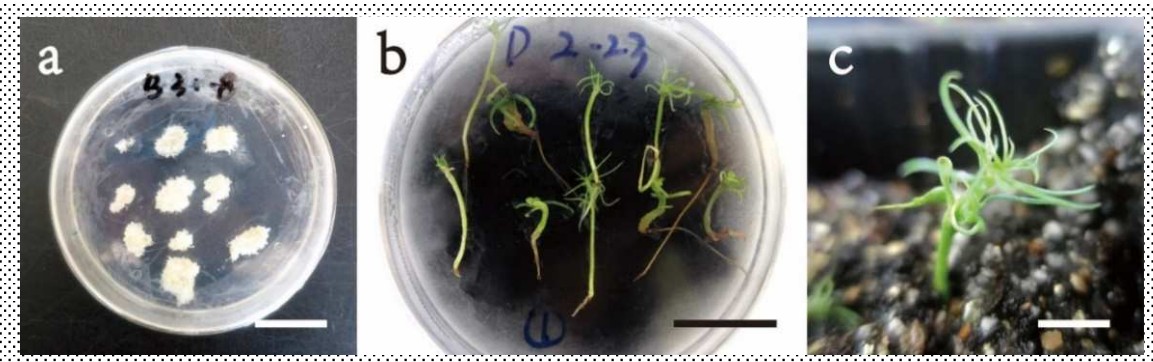

**Figure 8.** Recovery culture and seedling development in *Picea pungens* SE: (**a**) proliferated embryogenesis tissues recovered from cryopreserved tissues which were pre-treated with 0.4 mol/L sorbitol for 24 h and 5.0% DMSO; (**b**) embryo germination; and (**c**) growing embryo seedling. Bars: (**a**) = 4 cm, (**b**) = 4 cm, (**c**) = 2 cm.

## 4. Discussion

Low efficiencies in the SE initiation and maturation [1,4,5] as well as lacking an effective cryopreservation technology are the key barriers in integrating the SE technology into large-scale seedling production for *Picea pungens*. In this study, efforts were made, not so much to explore the underlying mechanisms controlling the SE processes, but to overcome the barriers via modifying media. In terms of efficiency, satisfactory results were achieved in all the areas.

### 4.1. Initiation of Embryogenic Tissues

Initiation frequency of *Picea pungens* SE could be changed by varying concentration of 2, 4-D or 6-BA, the most widely used PGRs in SE induction [1,4,5]; (Table 2). 2,4-D stimulates cell division and differentiation, playing a decisive role in initiation of embryogenesis [18]. The pattern of "rise-peak-drop" in initiation in response to an increasing dose of 2,4-D suggests that 2,4-D is essential but can be detrimental when overdosed. A too-high concentration of 2,4-D disrupted normal genetic and physiological processes in cells treated with this PGR, lowering intracellular contents of reduced glutathione, ascorbic acid and then the oxidative stress level (Unpublished data), which is unconducive to cell division. Based on this study, the optimal 2,4-D concentration was 4 mg/L (=36 μM), which was 40% more efficient than other concentrations tested. The recommended 2,4-D concentration in *Picea pungens* SE initiation varied, with some suggesting a low (9 to 10.0 μM) [1,5] and others a high (5.0 mg/L) concentration [4]. Virtually 6-BA acted on initiation in the similar pattern as 2,4-D did (Figure 1b). The optimal 6-BA concentration appeared at 1 to 2 mg/L (=44 μM), which is higher than those (5 to 10 μM) used by [1,3] but lower than that (5 mg/L) reported by [4]. It appears that a wide range of concentration for 2,4-D and 6-BA could be effective, but concentrations at 4 mg/L and 2 mg/L for 2,4-D and 6-BA, respectively, were favored based on our results. For spruce SE initiation, the concentration ranges from 0.4 to 2 mg/L for 2,4-D and 0.1 to 1 mg/L for 6-BA are recommended [6,19], and our results supported the higher concentrations for *Picea pungens*. Pine species favor even lower concentrations of 2,4-D and 6-BA; in *Pinus strobus*, the initiation frequency increased from an average of 0.20 to 0.53 by reducing the 2,4-D and BA concentrations from 9.5 to 2.2 μM and from 4.5 to 2.2 μM, respectively [14].

Sucrose plays duo roles in SE initiation as a carbon/energy source and as an osmotic regulator [20]. Varying its concentration would expect to impact initiation efficiency greatly, which was confirmed in this study (Figure 1c). The 10 g/L concentration obtained the highest initiation efficiency, but a lower concentration, 5g/L, achieved a statistically similar efficient. Our optimal concentration range of 5 to 10 mg/L goes along with findings in previous studies in *Picea pungens*, *Picea abies* and *Picea maiana* [5,6,20] but is at the low end of the general recommendation for conifer SE, where the use of 1%–3% sucrose was deemed optimal [21]. It seems the recommended sucrose levels supplemented to initiation media are accordant, around 10 g/L, at least for spruce species.

All provenances responded in initiation, but the efficiency varied (Table 3). A similar finding was reported in a study on *Picea pungens* [1], where all five provenances developed ETs with various efficiencies. Thus, for this species, the wide application of the initiation protocol is plausible, but low efficiencies may occur for some provenances. Clearly, provenances reacted to the treatments inconsistently (Table 3), i.e., relative to F2, F1 initiated better in Exp. 1.1 but poorer in Exp. 1.3. Note that provenance variation in this study might be underestimated due to the limited number of provenances being tested, and studies involving more genotypes are needed to achieve a firm conclusion. Despite the substantial variation, the interactions between provenance and treatment were negligible (Table 2), which is in parallel with the finding by Tao [1], suggesting the wide application of the identified optimum concentrations for the species.

A sufficient initiation is necessary for a practical SE program. Combining all provenances, this study achieved satisfactory initiation efficiencies (>0.50) and thorough judicial selections of concentrations of 2,4-D, 6-BA and sucrose (Figure 1), yielding an almost

two-fold increase compared to the average initiation frequency from 0.2 to 0.3 reported in previous studies [1,3–5]. Our efficiencies have attained to the levels reported in *Picea abies* (50%) and *Picea glauca* (30%–50%) [6,22,23], the most successful SE programs. Further improvement in the SE initiation for the species is possible, from a practical viewpoint, the achieved efficiencies in the present study are sufficient for a SE program.

*4.2. Maturation of Embryogenic Tissues*

Maturation efficiency in *Picea pungens* SE was improved by selecting a suitable basal medium of the maturation medium (Figure 4a). Previous studies on *Picea pungens* SE maturation have often utilized $\frac{1}{2}$MS as the basal medium [1,4]. Our results support the use of DCR, yielding 75% more mature embryos than that from using $\frac{1}{2}$MS or mLV (Figure 4a). The superiority of DCR over the other two may partly due to its lower concentrations of $NH_4^+$ and $NO_3^-$ and higher $Ca_2^+$ [24,25].

Suitable osmotic stress and water availability of the maturation medium are crucial to SE maturation and therefore the options for carbohydrate sources and their concentrations supplemented to the medium should be carefully weighted. Among the three carbohydrate types tested across a gradient of concentrations, the average efficiencies, while statistically comparable (Table 4), varied substantial, with sucrose yielding 47% more embryos than glucose, which in turn achieving 185% more mature embryos than maltose. Thus, the biological significance of the variation among the carbohydrate types might be greater than statistically evident, thereby emphasizing the importance of the choice of carbohydrate type. Clearly, for *Picea pungens* SE maturation, maltose was the least favorite, but between sucrose and glucose the choice was truly genotype-dependent (Figure 5a). Contradictory to our findings, sucrose was found detrimental to *Picea glauca* maturation [26] and in *Abies*, maltose, irrespective of concentration, was superior to sucrose [27]. Thus, optimal choices of carbohydrate sources might be species-dependent. Other than as an osmoticum, carbohydrates may influence maturation via various paths such as changing tissue's endogenous ABA [26,28] or starch accumulation [29]. Usually in the somatic maturation stage, higher concentrations of carbohydrate are essential to inhibit the premature germination of embryos, but too high concentrations may impede somatic embryo development. Our results supported the use of 30 g/L, regardless of carbohydrate type, for *Picea pungens* SE maturation, which concurs with the findings in other spruce species [29,30]. Exceptions do exist; a low concentration of 17 mg/L sucrose was found ideal for *Picea omorika* ET maturation [29].

Another factor pertinent to medium ostomic potential is the concentration of gelling agent, such as Gelrite, in the medium. A suitable concentration of Gelrite is crucial since it provides appropriate support for ETs in the medium to avoid clumping and hyperhydricity as well as improves nutrient diffusion. In this regard, while optimizing the Gelrite concentration for a wide application was not success since it was genotype-dependent (Figure 5b), our results do suggest the use of Gelrite with a concentration of less than 8 g/L (Figure 4c). High Gelrite concentrations (i.e., $\geq$8 g/L) limit the transport of sucrose from the medium to embryos and reduce starch levels in developing embryos, resulting in a detrimental effect on maturation in spruce [29]. This may not be true for Pinus species, of which a higher gel strength (9–12 g/L Gelrite) was favored [31,32]. Interesting, this study showed an increase in efficiency when the concentration increased from 8 to 10 mg/L for one line (2–23) (Figure 5c), which deserves further investigation.

To date, ABA has been primarily used in SE studies to promote maturation and synthesis of storage reserves in embryos during maturation [28,33]. Addition of exogenous ABA to the medium affects maturation efficiency, however, as regards to optimal concentration, existing literature is inconclusive. Low (8~12 uM) ABA concentrations in *Picea glauca*, *Picea mariana*, and *Picea omorika* [6,29,34], but higher (30~60 uM) concentrations to allow normal development and inhibit precocious germination in *Picea abies* and *Picea rubens* [8,11,35] have been recommended. Our results support the use of a high (10 g/L; 38 μM) concentration for *Picea pungens* (Figure 5c). Even higher ABA doses may be needed for pines;

e.g., in *Pinus koraiensis*, Peng [31] recommended to increase ABA concentration to 80 µM to maximize yield of cotyledonary somatic embryo.

Even the effect of culture density on maturation was not statistically significant (Table 4), the actual differences between the levels were substantial (Figure 4e), suggesting the practical importance of culture density on maturation. Clearly the optimal density varied with clone line and had to be matched by clone (Figure 5d), but culturing 110 mg or more fresh ETs per dish could have a negative impact on maturation. In the literature, few studies have investigated the topic. A 50 mg ETs per dish was recommended for maturation in Korean pine [31], which agrees well with the choice for one clone line in this study (Figure 5d).

All clone lines matured successfully but the efficiency varied (Table 2), supporting the findings in previous studies on *Picea pungens* [1] and other spruces [36]. Genotypes varied in response to the experiment conditions, e.g., while line 2–23 had a wide adaption to the conditions, yielding the highest efficiencies across all experiments, line 2–22 was sensitive to the treatments, being poor in Exp. 2.2 and 2.5 (Table 5). Different from initiation where the optimal choices did not vary with genotypes, most maturation experiments showed significant interactions between genotype and treatment (Table 4; Figure 5), and therefore, the optimal choices have to be matched by genotype. This might be a discouraging for commercializing SE technology for timber species where many genotypes are often involved in establishing plantations [37]. Such an impact would be at a much lesser level for ornamental species, such as *Picea pungens*, of which high-quality but a limited number of genotypes are often involved and a high price can always be commanded.

In this study, the obtained number of mature embryos ranged from 356 (Exp. 2.3) to 747 E/gFM (Exp. 2.1), representing a more than two-fold increase compared with the best results achieved in the previous studies, where the highest efficiency was 188 E/gFM [1,4]. Our maturation efficiencies go along with the greatest successes achieved in other spruce species. For example, Hazubska-Przyby [29] achieved 506 E/gFM in *Picea abies* and 490 E/gFM in *Picea omorika*, and Tikkinen [8] achieved 296 E/gFM in *Picea abies*. Thus, even the conditions for maturation can be further modified, the achieved efficiencies in the present should be sufficient for a practical SE program.

### 4.3. Cryopreservation of Embryogenic Tissues

Avoiding the loss of potential in embryo maturation during the long-term cryopreservation is key. In this study, the cryopreserved ETs were thawed, proliferated, and cultured, and the number of mature embryos counted to reflect the recovery efficiency from cryopreservation. This is different from previous studies where the maturation efficiency was indirectly evaluated by vital staining methods such as TTC and FDA [31].

This study demonstrated that optimizing preculturing conditions was a critical step in the successful cryostorage of somatic tissues (Table 4). Toward this end, our results confirmed that the optimal sorbitol concentration interacted with the preculture period (Table 4; Figure 7), with the best combination being preculturing 24 h with 0.4 mol/L sorbitol. A statistically comparable efficiency was also obtained by the combination of a higher sorbitol concentration of 0.8 mol/L and a shorter preculture period of 12 h. Optimal combinations for sorbitol and preculture period have been reported, e.g., 0.8 mol/L sorbitol with preculturing 48 h in *Picea mariana* [38] and 0.4 mol/L sorbitol with preculturing 18 h in Korean pine [31]. It seems a sorbitol concentration from 0.4 to 0.8 mol/L is suitable but the preculture period can vary greatly, depending on species. After the preculture of ETs with sorbitol, the ETs are often treated with DMSO, a commonly used cryoprotectant, and our results supported the use of 5% DMSO (Figure 7b, Figure 8a), which agrees well with findings in other spruce species [39,40]. The cryotoxicity of DMSO to cells is well-known [41], which, however, can be offset when the tissues are pre-treated with a suitable level of sorbitol [42]. Optimal combinations of preculturing 20 to 24 h with 0.4 mol/L sorbitol, followed by treating with 33% DMSO for *Picea mariana* [6], and preculturing 18 h with 0.4 mol/L sorbitol and then treating with 7.5% DMSO for *Pinus pinaster* [43] have been

reported. Based on this study, a preculture procedure of culturing 24 h with 0.4 mol/L sorbitol, followed by treating with 5% DMSO is recommended for *Picea pungens*.

Overall, the high maturation efficiencies (1030 E/gFM in Exp. 3.1 and 792 E/gFM in Exp. 3.2) from culturing cryopreserved ETs in this study suggest that ETs can be stored in liquid nitrogen in the juvenile stage, without losing the viability of the cryopreserved tissues. Culturing ET clumps on medium supplemented with 0.4 M sorbitol for 24 h and then 5% DMSO prior to cryopreservation was effective in inducing dehydration and freezing tolerance. In addition to maintain regenerative capacity, maintaining tissue's genetic fidelity is indispensable for a cryopreservation technique, which was not investigated in the present study. Clearly the recovery efficiency from a longer (i.e., $\geq 5$ years) storage and involving more genotypes should be investigated in future. Furthermore, vitrification-based cryopreservation methods for long-term storage of mature somatic embryos of *Picea* spp., where cryopreservation of pre-dried somatic embryos could be achieved without the use of cryoprotectants, have been demonstrated to be effective [44,45]. Dried and cryopreserved somatic embryos of *Picea* could be used to reinduce embryogenic tissue and start new embryo production [44]. Compared to the method used in this study, the vitrification-based methods also have the advantages in cost and simplicity, and their applications to cryopreserve ETs of *Picea pungens* deserve further exploration. Nonetheless, in terms of recovery rate, the preculturing conditions for cryopreservation defined in the present study are sufficient to support a practical SE program for the species.

## 5. Conclusions

Two key efficiency-limiting, early steps in *Picea pungens* SE, initiation and maturation, were the obstacles for incorporating the technology into large-scale seedling production programs. This study greatly improved initiation efficiency from around 0.25 in frequency reported in previous studies to 0.56 by judicious selections of 2,4-D, 6-BA and sucrose concentrations. The genotype by treatment interactions was negligible in the initiation experiments, and therefore, the optimal concentrations (4 mg/L 2,4-D, 2 mg/L for 6-BA, and 5 to 10 mg/L sucrose) could be appropriate for a wide range of genotypes. The study enhanced maturation efficiency from maturing as high as 188 E/gFM in the previous studies to 743 E/gFM via modifying the maturation media. Culturing ETs on the maturation medium using DCR as the basal medium or supplemented with 10 mg/L ABA was favored in terms of improving maturation. Adjusting the osmotic stress (via adjusting carbohydrates and Gelrite) of the maturation medium and the culture density were also effective in improving maturation efficiency, but the optimal selections were genotype-dependent and had to be properly matched to genotype. Moreover, the present study optimized the preculturing conditions for ET cryopreservation. Preculturing fresh ETs 24 h with 0.4 mol/L sorbitol and then treating the ETs with 5% DMSO was the recommended pretreatment procedure. While the study objectives were fulfilled, further studies are needed, in particular on (1) somaclonal variation and recovery capability from long-term cryopreserved (i.e., >5 years) tissues, (2) other cost-efficient cryopreservation methods such as vitrification-based ones, (3) field performance of SE seedlings and (4) automation of SE steps to reduce cost. Overall, the results of this study were encouraging, enhancing efficiencies to the levels appropriate for commercializing SE technology. Results of this study, together with the well-refined protocols for other SE steps [1], suggest a considerable promise for vegetative propagation using SE in *Picea pungens*, which, if integrated into the current genetic selection program, would potentially make this species more attractive.

**Author Contributions:** Conception, X.C., F.G. and J.T., writing-original draft preparation, X.C., J.T., F.G. and Y.W.; writing-review and editing, X.C., F.G., J.T. and Y.W.; experimental planning, X.C., F.G. and J.T.; experiment set up and data collection, X.C., F.G., C.Q., S.C., J.C., C.S.; data analysis, X.C., Y.W. All authors have read and agreed to the published version of the manuscript.

**Funding:** This research is supported by the Major Science and Technology Project "Research on Collection and Preservation Technologies of Germplasm Resources in the Trees of Changbai Moun-

tain" of Jilin Provincial Forestry (2015-002), the Forestry Science and Technology Promotion and Demonstration Project, "Promotion of Efficient Propagation Technologies in Blue Spruce" funded by the Central Finance of China (No. JLT2021-10).

**Data Availability Statement:** The data is available on request from the corresponding author.

**Conflicts of Interest:** The authors declare no conflict of interest.

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
