# Peer review of "Optimizing Somatic Embryogenesis Initiation, Maturation and Preculturing for Cryopreservation in Picea pungens"

_forests, doi:10.3390/f13122097_

Round 1
Reviewer 1 Report
Thanks for submitting an article to this journal.
It would help if you revised the entire article.
Correct spelling mistakes.
Correct grammar mistakes.
About Somatic Embryogenesis, Reinforce the introduction and discussion sections and refer to other related references such as:
Keshvari, T., Najaphy, A., Kahrizi, D., & Zebarjadi, A. (2018). Callus induction and somatic embryogenesis in Stevia rebaudiana Bertoni as a medicinal plant. Cellular and Molecular Biology, 64(2), 46–49. https://doi.org/10.14715/cmb/2018.64.2.9
Taimori, N., Kahrizi, D., Abdossi, V., & Papzan, A. H. (2016). Cell dedifferentiation, callus induction and somatic embryogenesis in Crataegus spp. Cellular and Molecular Biology, 62(11), 100–107. https://doi.org/10.14715/cmb/2016.62.11.17
Pramanik, S., Williams, A., & Bewley, J. (2007). In vitro reconstitution of legumin (11s) mRNA and binding proteins as related to post-transcriptional regulation of protein synthesis in developing alfalfa embryos. Cellular and Molecular Biology, 53(3), 64–73. Retrieved from https://www.cellmolbiol.org/index.php/CMB/article/view/1128
Author Response
Thanks for submitting an article to this journal.
It would help if you revised the entire article.
Correct spelling mistakes.
Correct grammar mistakes.
We greatly appreciate the suggestions . We revised the manuscript thoroughly to focus more on results. For grammar and spelling, we tried our best to correct. One of my graduate students (native English speaker) helped with this too.
About Somatic Embryogenesis, Reinforce the introduction and discussion sections and refer to other related references such as:
Keshvari, T., Najaphy, A., Kahrizi, D., & Zebarjadi, A. (2018). Callus induction and somatic embryogenesis in Stevia rebaudiana Bertoni as a medicinal plant. Cellular and Molecular Biology, 64(2), 46–49. https://doi.org/10.14715/cmb/2018.64.2.9
Taimori, N., Kahrizi, D., Abdossi, V., & Papzan, A. H. (2016). Cell dedifferentiation, callus induction and somatic embryogenesis in Crataegus spp. Cellular and Molecular Biology, 62(11), 100–107. https://doi.org/10.14715/cmb/2016.62.11.17
Pramanik, S., Williams, A., & Bewley, J. (2007). In vitro reconstitution of legumin (11s) mRNA and binding proteins as related to post-transcriptional regulation of protein synthesis in developing alfalfa embryos. Cellular and Molecular Biology, 53(3), 64–73. Retrieved from https://www.cellmolbiol.org/index.php/CMB/article/view/1128
We enjoyed reading all these publications, very helpful in understanding roles of PGRs. There are too many publications on the topic of various species. We decided to focus more on conifers in particular spruce species in the introduction and discussion, which is more relevant to this study. Following the suggestions, we revised both sections thoroughly.
Reviewer 2 Report
Optimizing Somatic Embryogenesis Initiation, Maturation, and Preculturing for Cryopreservation in Picea pungens
Comments and Suggestions for Authors
Abstract:
Lines 26 write the value of osmolarity of your maturation medium.
Line 27 write an exact concentration of gelrite in a maturation medium, this sentence is confusing.
Introduction:
Well written.
M and M
Line 111, please make sure that her mentioning long/term seeds temperature was really minus 40 ͦC. sounds unusually low for seed storage.
Line 198 please write the full name and manufacturer specification of the programmed cooling box.
Results:
Well written, but there are absolutely no pictures of the embryogenic cultures in different stages of the technology: initiation, maturation, SE, and callus used for the cryopreservation !!
Line 296 DCR medium composition or reference is missing.
Fig. 2 b is very bad quality, and not legible, please improve the resolution. An explanation of the abbreviation used is missing in all Figures.
Fig. 3 add an explanation of what is SE/g FM.
Fig. 4 add an explanation of what is SE/gFW.
Discussion:
Lines 439 and 440 1/2 MS medium and DCR medium reference are missing.
Abbreviations: this section is missing; please add it to the manuscript. e.g. DMSO, DCR medium, 1/2 MS, etc.
Revision 20.11.2022

Author Response
Comments and Suggestions for Authors
Abstract:
Lines 26 write the value of osmolarity of your maturation medium.
We did not measure the osmolarity. We clarified this by adding (See line 26):
“by adjusting the maturation medium’s osmotic pressure by manipulating the concentrations of carbohydrate and Gelrite”
Line 27 write an exact concentration of gelrite in a maturation medium, this sentence is confusing.
The optimal Gelrite concentration was genotype-dependent, so no exact value can be provided. We moved ‘(<8 g/L)’ ahead to clarify the sentence. See line 28.
Introduction: Well written.
M and M
Line 111, please make sure that her mentioning long/term seeds temperature was really minus 40 ͦC. sounds unusually low for seed storage.
Good comment. The actually T was -20C. Corrected it. See line 107.
Line 198 please write the full name and manufacturer specification of the programmed cooling box.
Relevant information was provided. See line 191.
Results: Well written, but there are absolutely no pictures of the embryogenic cultures in different stages of the technology: initiation, maturation, SE, and callus used for the cryopreservation !!
Good suggestions and we agree. See responses to Editor’s comments.
Line 296 DCR medium composition or reference is missing.
Reference for DCR could be found on line 167.
Fig. 2 b is very bad quality, and not legible, please improve the resolution. An explanation of the abbreviation used is missing in all Figures.
A new Fig. 2b was provided. Abbreviations were provided. See line 345.
In addition, changes were made to all figures with bars so that the standard errors can be seen better.
Fig. 3 add an explanation of what is SE/g FM.
To be consistent with the text, ‘SE/gFM’ was changed to ‘E/gFM’ in all figures. An explanation was added.
Fig. 4 add an explanation of what is SE/gFW.
An explanation of E/gFM was added.
Discussion:
Lines 439 and 440 1/2 MS medium and DCR medium reference are missing.
References for ½ MS and DCR can be found in line 167.
Abbreviations: this section is missing; please add it to the manuscript. e.g. DMSO, DCR medium, 1/2 MS, etc.
See line 89 for DMSO, and line 167 for DCR and ½ MS.
Reviewer 3 Report
Respected authors are requested to answer the following questions and improve the paper by adding below recommendations:
Figures related to different experimental steps (embryo isolated from the seeds, SE tissues in different conditions, etc) should be added.
Line 116-117,
Were the seeds in the same age and embryos were in the same size? add pictures from isolated embryos.
Line 158,
which filter paper was used ?
Line 169,
Figure showing mature embryo should be added.
Line 195,
what's the light photoperiod?
Line 292,
what's the reason of introduce 10 g/L as optimal conc., while there is no significant difference between 10 g/L and 5 g/L?
Line 310,
Figure 2b or 3b? It seems 2b. correct it.
Line 330,
How do you explain the lack of significant difference?
Line 363,
Add pictures from recovered mature embryo vs failed mature.
Line 392,
Based on literature, add the reason of 2,4-D high level inhibition.
Line 409,
what's the reason for select 10 g/L sucrose as an optimal concentration while there is no significant between 5 and 10 ?
Line 414-415,
how do you explain this discrepancy in molecular-physiological level ? please add some explanation.
Author Response
Figures related to different experimental steps (embryo isolated from the seeds, SE tissues in different conditions, etc) should be added.
Good suggestions. See responses to Editor’s comments. We do have pictures of isolated embryos. We did not include them to avoid too lengthy of the manuscript.
Line 116-117, Were the seeds in the same age and embryos were in the same size? add pictures from isolated embryos.
Seed sources information can be found in lines 107-109. We did not measure sizes of embryos, so no information can be provided.
Line 158, which filter paper was used ?
Information was added. See line 152
Line 169, Figure showing mature embryo should be added.
See above. Provided. See line 371
Line 195, what's the light photoperiod?
Information was added. See line 188.
Line 292, what's the reason of introduce 10 g/L as optimal conc., while there is no significant difference between 10 g/L and 5 g/L?
This is a good comment. Statistically, there were the same. We made changes to reflect this.
Lines 296-297.
“Overall, a sucrose concentration from 5 to 10 g/L was ideal for the SE induction of Picea pungens, which was not changed with provenance.”
Also see changes in Abstract (line 21), discussion (lines 442-446) and conclusion (line 593)
Line 310, Figure 2b or 3b? It seems 2b. correct it.
We checked this carefully and changed it to Figure 5c (line 329). The original Figure 2b was replaced by a new one.
Line 330, How do you explain the lack of significant difference?
This is a difficult statistical question. The main reason for the lack of significant difference, like the other experiments (see line 306), is the large standard errors associated with the averages, which in turn are related to sample size. While statistically insignificant for a difference, the biological importance of the difference cannot be ignored, which was emphasized in the discussion. See lines 477-482 and lines 517-519.
Line 363, Add pictures from recovered mature embryo vs failed mature.
We added a picture to show recovered tissues and developed seedlings. New Figure 8. Line 406.
Line 392, Based on literature, add the reason of 2,4-D high level inhibition.
This is a great suggestion. Actually, we have unpublished data to answer the question. See lines 423-426.
“Too high concentration of 2,4-D disrupted normal genetic and physiological processes in cells treated with this PGR, lowering intracellular contents of reduced glutathione, ascorbic acid, and then the oxidative stress level (Unpublished data), which is unconducive to cell division.”
Line 409, what's the reason for select 10 g/L sucrose as an optimal concentration while there is no significant between 5 and 10 ?
We made changes to reflect that the 5 and 10 mg/L were statistically the same.
Lines 296-297.
“Overall, a sucrose concentration from 5 to 10 g/L was ideal for the SE induction of Picea pungens, which was not changed with provenance.”
Line 414-415,
how do you explain this discrepancy in molecular-physiological level? please add some explanation.
In the original version, lines 414-415 were about the optimal concentration of sucrose in initiation media as below:
“It seems the recommended sucrose levels supplemented to initiation media are accordant, around 10 g/L, at least for spruce species SE.”
No discrepancy was described. For personal interesting, I looked into the topic but could not find publications related to sucrose effects on initiation in molecular-physiological level.
Round 2
Reviewer 2 Report
Optimizing Somatic Embryogenesis Initiation, Maturation, and Preculturing for Cryopreservation in Picea pungens
Comments and Suggestions for Authors
Unfortunately, the pictures added to the manuscript are of very bad quality, especially Fig.1. In all pictures the scale bar is missing. This can not be published like it is now.
28.11.2022
Author Response
Unfortunately, the pictures added to the manuscript are of very bad quality, especially Fig.1. In all pictures the scale bar is missing. This can not be published like it is now.
Responses:
We replaced some of the pictures including figure 1. We also added bar scales for each picture.